# Relationship between Stock Returns and Trading Volume at the Bourse Régionale des Valeurs Mobilières, West Africa

**Jean-Pierre Gueyie [1], Mouhamadou Saliou Diallo [2,*] and Mamadou Fadel Diallo [2]**

1    Department of Finance, School of Management, University of Quebec in Montreal, Montreal, QC H2X 3X2, Canada
2    Faculty of Economics and Management, Cheikh Anta Diop University, Dakar 4163, Senegal
*    Correspondence: mouhamadousaliou.diallo@ucad.edu.sn

**Abstract:** The objective of this paper is to study the contemporaneous relationship and the dynamic relationship between the stock index return and the trading volume on the Bourse Régionale des Valeurs Mobilières using daily data from 5 January 2015 to 31 October 2022. Estimations are made using the generalized method of moments (GMM) and generalized autoregressive conditional heteroscedasticity or GARCH (1,1) specifications for the contemporaneous regressions and the vector autoregressive specification for the dynamic (causal) relationship. The contemporaneous specifications show that there is no significant relationship between stock returns and trading volume. Neither of the two variables significantly influences the other. Furthermore, the dynamic specification shows a causality running from stock returns to trading volume but the reverse is not true. For the period covered by the study, the results invalidate both the mixture of distribution hypothesis and the sequential information arrival hypothesis and open the way for other considerations such as behavioral models.

**Keywords:** stock returns; trading volume; dynamic relationship; Granger causality; Bourse Régionale des Valeurs Mobilières (BRVM); West Africa

## 1. Introduction

The microstructure theory seeks to explain the process of price formation in financial markets (O'Hara 1995). It suggests that both price changes (i.e., returns) and trading volume are related to the arrival of information to the market. Thus, price movement (return) and trading volume jointly depend on the intensity of information flow (Wang et al. 2018).

In the literature, several theoretical propositions (hypotheses) link stock returns and trading volume or volatility and trading volume. The two most cited hypotheses will be considered in this paper: (i) the mixture of distribution hypothesis [MDH] and (ii) the sequential information arrival hypothesis [SIAH] (Lee and Rui 2002; Habib 2011; Akpansung and Gidigbi 2015; Naik et al. 2018; Nyakurukwa 2021; Ngene and Mungai 2022).

The mixture of distribution hypothesis [MDH] (Clark 1973; Epps and Epps 1976) postulates that daily price changes (returns) are driven by information flow. It uses trading volume to measure disagreement among traders as investors revise their market prices based on the arrival of new information on the market. The trading volume level increases as the degree of disagreement among traders widens. Stock returns and trading volume are related due to their common dependence on a latent information flow variable. The MDH then suggests a positive contemporaneous relationship between trading volume and stock returns.

The sequential information arrival hypothesis [SIAH] states that when new information reaches the marketplace, it is not transmitted to all investors simultaneously (Copeland 1976; Jennings et al. 1981; Jennings and Barry 1983; Smirlock and Starks 1988). Instead, it is disseminated sequentially, reaching the marketplace at different times and speeds.

Incomplete equilibriums are generated, leading to a final informational equilibrium when all participants hold the information. Thus, adjustment to new information is not immediate (Naik et al. 2018). The SIAH advocates the existence of a reciprocal and positive relationship between trading volume and stock price change (returns). Lagged trading volume may contain information that is useful in predicting current stock returns. In the same vein, lagged stock returns may also contain information that is useful in predicting current trading volume.

These two hypotheses, MDH and SIAH serve as the basis for the theoretical link that is developed between returns and trading volume.

Regarding the empirical setting, extensive literature exists on the relationship between stock returns and trading volume (see Karpoff 1987; Mahajan and Singh 2009). However, most of the studies involve developed stock markets. Empirical studies performed on African stock exchanges are limited in number. Adding to this, most of the existing studies deal with the relationship between returns volatility and trading volume. No prior research has been conducted on this topic with regard to West African Economic and Monetary Unit (WAEMU) countries. Accordingly, there is interest in studying the relationship between these two variables in the context of the Bourse Régionale des Valeurs Mobilières (BRVM), a regional stock exchange located in Abidjan, Ivory Coast).[1]

Three research questions are addressed in this paper: 1. What is the contemporaneous relationship between stock returns and trading volume at BRVM? 2. Does a causality exist between stock returns and trading volume at BRVM? 3. If so, what is its direction? Thus, the objective of the paper is to shed light on the relationship between stock returns and the volume of securities traded by answering these questions. Analyses are conducted in both contemporaneous and dynamic contexts.

Our results show that there is no significant contemporaneous relationship between stock returns and volume. Neither of the two variables influences the other. The dynamic specification shows a causality running from stock returns to trading volume, but the reverse is not true.

The main contribution of this research is its focus on BRVM, a market that is unexplored in the literature as far as the relationship between stock returns and trading volume is concerned. This stock exchange is of particular interest because it is common to eight WAEMU countries.

The paper continues as follows: Section 2 presents a brief literature review, Section 3 outlines the data and methodology, Section 4 reports and discusses the empirical results, and Section 5 concludes the paper.

## 2. Literature Review

As stated previously, several theoretical propositions in the literature link stock returns and trading volume or returns volatility and trading volume. The hypotheses MDH (Clark 1973; Epps and Epps 1976) and SIAH (Copeland 1976; Jennings et al. 1981; Jennings and Barry 1983; Smirlock and Starks 1988) were summarized in the introduction. They serve as the basis for the theoretical link that is developed between returns and trading volume.[2] Other theoretical contributions can be found in De Long et al. (1990), Campbell et al. (1993), Wang (1994) and He and Wang (1995). Ngene and Mungai (2022) have also summarized the dispersion of beliefs hypothesis [DBH] of Shalen (1993) and Harris and Raviv (1993) and the information based hypothesis [IBH] of Blume et al. (1994).

The theoretical development has been paving the way for a significant body of empirical literature using different data and contexts. Earlier studies have focused on the contemporaneous relationship between stock returns and trading volume (Epps and Epps 1976; Tauchen and Pitts 1983; Wood et al. 1985; Gallant et al. 1992). Interesting literature reviews can be found in Karpoff (1987), Mahajan and Singh (2009) and Akpansung and Gidigbi (2015). These earlier researches have been extended with attention given to the dynamic side of the relationship between stock returns and trading volume.

In the developed country setting, Lee and Rui (2002) established that trading volume is Granger-caused by stock returns on the developed markets of the US, the UK and Japan, while the reverse is not true (trading volume does not Granger-cause stock market returns on each of the three stock markets). Furthermore, cross-country analyses showed that US trading volume has a predictive power for the financial markets in the UK and Japan. Cook and Waston (2017) used data from the FTSE 100 index in the UK over the period January 1991 to July 2016 and the daily opening, closing, and high and low values to analyze the relation between trading volume and stock returns. They found a mixed and complex picture of causality in the returns-volume relationship. Specifically, they found significant bidirectional causality when returns are calculated on the basis of the daily opening and high and low values of the FTSE 100 index. The results for returns based on the daily closing values provide evidence of unidirectional causality from returns to volume. Cook and Watson also performed analyses using rolling regressions to account for time variability. Overall, they reported that the returns-volume relationship is not a single entity to be assessed and classified. It varies across the alternative measures of return that are available, and it varies over time. Wang et al. (2018) studied the dynamic relation between trading volume and stock returns from the perspective of out-of-sample stock return predictability using both USA and international (G-7 countries) data. Their aim was to assess whether there is any relation between the two variables, and if the answer is yes, whether such a relation is economically significant. First, they found that trading volume consistently helps predict future return but only for the equal-weighted portfolio (no effect is found for value-weighted ones), suggesting that this predictive power of trading volume is mainly driven by small stocks. The economic profits of the ex-ante (out-of-sample) predictability is small. Accordingly, investors are not likely to gain financially from their transactions.

For emerging markets (excluding Africa), Pisedtasalasai and Gunasekarage (2007) tested the causal and dynamic relationship among returns, return volatility and trading volume for the equity markets of Indonesia, Malaysia, the Philippines, Singapore and Thailand. They found a strong asymmetric relationship between stock returns and trading volume. Specifically, returns are important in predicting their future dynamics as well as those of trading volume, while trading volume is found to have a limited impact on the future of the dynamics of stock returns. Chen (2008) explored the linear and non-linear causal relationship between stock price and trading volume in China from 1993 to 2006 for the Shanghai A share and from 1994 to 2006 for the Shanghai B share. Using the Auto Regressive Distributed Lag (ADRL) methodology, the Granger causality test and the non-Granger causality test, Chen reported several results: from ADRL, a long-run level equilibrium relationship between the stock price and trading volume. From linear causality, Chen reported a unidirectional causality from price to volume in the case of Shanghai B and Shenzhen B shares in the short-run and a bidirectional causality between price and volume for the Shanghai A and Shenzhen A shares. For non-linear Granger causality, the researcher reported a neutral price-volume causal relation for the Shanghai B share and a bidirectional non-linear price-volume relation for the case of the Shanghai A and Shenzhen A shares. Finally, Chen reported a unidirectional non-linear causal relation running from stock price to trading volume for the case of the Shenzhen B share. In another study, Lin (2013) used the test of Granger non-causality in quantiles to investigate the dynamic relationship between returns and trading volume in six Asian countries (Indonesia, Malaysia, Singapore, South Korea, Taiwan and Thailand) using daily data from 1990 to 2008. The results show that in all Asian markets except Taiwan, trading volume Granger-causes stock returns in quantiles and the causal effects of volume are heterogeneous across quantiles. They have positive causal effects for upper quantiles and negative effects for lower quantiles. The effects are stronger at more extreme quantiles. Other emerging market studies include Pathirawasam (2015) [Sri Lanka]; De Medeiros and Doornik (2006) [Brazil]; and Tapa and Hussin (2016) [Malaysia].

In Africa, Abukari and Assogbavi (2019) used weekly data on 36 companies listed on the Johannesburg Stock Exchange (JSE) between 2011 and 2017 and reported both a

contemporaneous relationship and a bidirectional causal relationship between stock returns and trading volume. Habib (2011) investigated the dynamic (causal) relationship between stock returns and trading volume on the Egyptian securities exchange over the period 1998-2005 and found no relation between the lagged trading volume and the current stock returns. Mpofu (2012) used daily data from the FTSE/JSE index (South Africa) over the period 22 July 1988 to 11 June 2012 and reported that stock returns are positively related to the contemporary change in trading volume and unidirectional causality running from returns to volume. Akpansung and Gidigbi (2015) studied the Nigerian stock exchange over the period 1981 to 2012 and identified a long-run relationship between change in the trading volume and the returns. However, neither of the two variables Granger-causes the other. Ngene and Mungai (2022) used data on eight African stock markets (Egypt, Ghana, Kenya, Mauritius, Morocco, Nigeria, South Africa and Tunisia) to investigate the asymmetric and intertemporal causality among stock returns, trading volume and volatility. They found that stock returns generally Granger-cause trading volume, while lagged trading volume has a negative causal effect on returns at low quantiles and a positive causal effect on them at high quantiles. Toe and Ouedraogo (2022) considered eleven African stock exchanges over the period 24 September 2010 to 24 September 2020 (i.e., a total of 3037 daily observations per country). They found that returns do not cause volume, while volume causes returns in the stock exchanges of some countries. Nyakurukwa (2021) revisited the dynamic stock return-volume relationship in South Africa for the period 04 January 2005 to 26 February 2021, using a non-parametric causality in quantiles approach. First, using Granger causality tests on the mean, Nyakurukwa found a unidirectional causality from volume to returns in the stable periods (pre-2008 financial crisis and post-2008 financial crisis periods) and in the full sample. The in-crisis and post-COVID 19 periods were marked by bidirectional causality between the two series. Second, implementing a non-parametric quantile causality approach, Nyakurukwa found a causality from returns to trading volume in the middle quantiles of the conditional distributions during stable periods (pre-crisis and post-crisis) as well as the full sample: a causality that disappears during the in-crisis and the post-COVID 19 periods. Across all the samples used, no evidence was found of causality from trading volume to returns. The divergence of Granger causality tests in the mean and non-parametric quantile causality approach results allowed Nyakurukwa to point out the importance of going beyond the traditional Granger causality tests, which are solely based on conditional means.

As we can see, the number of empirical studies (relating trading volume and stock returns) performed on African stock exchanges is limited (Abukari and Assogbavi 2019; Habib 2011; Akpansung and Gidigbi 2015; Nyakurukwa 2021; Ngene and Mungai 2022). Other studies exist, but most of them deal with the relationship between trading volume and stock market volatility (Habib 2011; Naik et al. 2018).

Habib (2011) investigated the contemporaneous relationship between volume and volatility using the ordinary least squares (OLS) and generalized autoregressive conditional heteroscedasticity (GARCH) models, as well as the dynamic (causal) relationship between trading volume and volatility on the Egyptian securities exchange over the period 1998–2005. Habib first found that the lagged trading volume has a small role to play in forecasting the future return volatility. Furthermore, Granger causality tests indicate a bidirectional causal relation between volume and volatility. Naik et al. (2018) studied the South African stock exchange over the period 6 July 2006 to 31 August 2016. They found a positive contemporaneous relationship between returns and volatility and a unidirectional causality between volume and volatility, with trading volume Granger-causing market volatility.[3]

On the methodological side, several methods have been used, such as correlation analysis (Chordia and Swaminathan 2000); ordinary least squares (Habib 2011; Tapa and Hussin 2016); the generalized method of moments [GMM] (e.g., Lee and Rui 2002; Mougoué and Aggarwal 2011; Yonis 2013; Abdullahi et al. 2014); and GARCH methods (e.g., Lee and Rui 2002; Mahajan and Singh 2008; Mahajan and Singh 2009; Habib 2011; Yonis 2013; Komain 2016) for the contemporaneous relationship. The methods used also include

traditional Granger causality tests (e.g., Lee and Rui 2002; Habib 2011; Abdullahi et al. 2014; Akpansung and Gidigbi 2015; Nyakurukwa 2021; Ngene and Mungai 2022) and the non-parametric quantile causality approach (Nyakurukwa 2021).

From the theoretical and empirical literature review, we can formulate the following hypothesis:

**Hypothesis 1a.** *Trading volume has a positive contemporaneous effect on stock returns.*

**Hypothesis 1b.** *Stock returns have a positive contemporaneous effect on trading volume.*

**Hypothesis 2a.** *Trading volume Granger-causes stock returns.*

**Hypothesis 2b.** *Stock returns Granger-cause trading volume.*

Testing these hypotheses will shed additional light on a market that (as far as the topic treated is concerned) is unexplored in the literature, the BRVM. This is the main contribution of this paper. The hypotheses are tested using the following data and methodology.

## 3. Data and Methodology

### 3.1. Data and Preliminary Tests

The data used in this paper consist of the daily market price index and the daily trading volume series for the BRVM.[4] They are extracted from the BRVM financial database. They cover the period 5 January 2015 to 31 October 2022, which is a total of seven years, ten months or 1949 observations.

We have tested for a unit root in price (returns) and volume (logarithm of the volume) series using both the augmented Dickey and Fuller (1979) [DF] tests and the Phillips and Perron (1988) [PP] tests. In these tests, the null hypothesis is that a series is nonstationary. The results are reported in Table 1.

**Table 1.** Unit Root Tests.

| Variables | Augmented Dickey-Fuller Test | | Phillips-Perron Test | |
|---|---|---|---|---|
| | Level | First Difference [1] | Level | First Difference [2] |
| Price | −1.014 | −16.40 *** [a] | −0.853 | −45.43 *** [a] |
| Volume | −15.18 *** | −10.09 *** [b] | −40.10 *** | −34.20 *** [b] |

Notes. *** indicates significance at the 1% level; [a] indicates returns instead of first difference; and [b] indicates logarithm of volume instead of first difference. [1] Series that are not stationary in level usually become stationary after a certain number of differentiations (usually one). In this table, prices have been differentiated, but the resulting changes in price are expressed in percentage (which means return). For the volume series, instead of differentiating, we have used the log transformation, which also allows stationarity. [2] see Note 4.

From the results of both the augmented Dickey-Fuller test and the Phillips-Perron test, the price series is not stationary in level but becomes stationary when differenced (i.e., expressed in returns here). Both the original volume and logarithm of volume series are stationary. After the requested transformation, the series are ready for the tests conducted within the paper.

### 3.2. Methodology

The analyses are conducted using both contemporaneous and dynamic contexts. All the models used in this paper are borrowed from previous researches on the relationship between stock returns and trading volume or between returns volatility and trading volume.

For the contemporaneous relationship estimation, we first use the generalized method of moments (GMM) specification. GMM has an advantage over the simple correlation or the simple ordinary least square (OLS) model because it accounts for autocorrelation, heteroscedasticity and endogeneity problems that are present in financial series. It has been

used by several authors in returns-volume/volatility-volume estimations (e.g., Lee and Rui 2002; Mougoué and Aggarwal 2011; Yonis 2013; Abdullahi et al. 2014). GMM is also used in the financial literature, for various regression estimations (Ali et al. 2020; Gueyie et al. 2019). It stands as follows:

- Contemporaneous GMM specification

$$\begin{aligned} R_t &= a_0 + a_1 V_t + a_2 R_{t-1} + a_3 V_{t-1} + \varepsilon_t \\ V_t &= b_0 + b_1 R_t + b_2 R_{t-1} + b_3 V_{t-1} + u_t \end{aligned} \tag{1}$$

where R is the return and V is the logarithm of the trading volume; $a_0$, $a_1$, $a_2$, $a_3$, $b_0$, $b_1$, $b_2$ and $b_3$, are coefficients to be estimated; and $\varepsilon_t$ and $u_t$ are the error terms.

We complement this GMM specification with a GARCH (1,1) specification.[5] A preliminary normality test (not reported here) shows that daily returns present some heteroscedasticity. This is accounted for in the estimation by the GARCH specification. GARCH family models have also been used by several authors in the literature in returns-volume/volatility-volume estimations (e.g., Lee and Rui 2002; Mahajan and Singh 2009; Habib 2011; Yonis 2013; Komain 2016). The GARCH (1,1) specification stands as follows:

- Contemporaneous GARCH (1,1) specification

$$R_t = a_0 + a_1 V_t + \varepsilon_t, \quad \varepsilon_t | (\varepsilon_{t-1}, \varepsilon_{t-2}, \ldots\ldots) \to N(0, h_t) \tag{2a}$$

$$h_t = \delta_0 + \delta_1 \varepsilon_{t-1}^2 + \delta_2 h_{t-1}$$

$$V_t = b_0 + b_1 R_t + u_t, \quad u_t | (u_{t-1}, u_{t-2}, \ldots\ldots) \to N(0, h_t) \tag{2b}$$

$$h_t = \lambda_0 + \lambda_1 \varepsilon_{t-1}^2 + \lambda_2 h_{t-1}$$

where R is the return and V is the logarithm of the trading volume, and $h_t$ represents the conditional variance term in period t.

- **Dynamic specification**

The dynamic specification uses a vector autoregressive regression (VAR) specification, which is widely considered in the literature when performing Granger causality tests (e.g., Lee and Rui 2002; Habib 2011; Abdullahi et al. 2014, Akpansung and Gidigbi 2015; Ngene and Mungai 2022). Lagged values of both R and V are considered to predict their current values. V Granger-causes R if the past values of V help predict the current value of R. Similarly, R Granger-causes V if the past values R help predict the current value of V. The VAR specifications stand as follows:

$$\begin{aligned} R_t &= \alpha_0 + \sum_{i=1}^{m} \alpha_i R_{t-i} + \sum_{i=1}^{n} \phi_i V_{t-i} + \varepsilon_t \\ V_t &= \gamma_0 + \sum_{i=1}^{m} \gamma_i R_{t-i} + \sum_{i=1}^{n} \varphi_i V_{t-i} + u_t \end{aligned} \tag{3}$$

where R is the return; V is the logarithm of the trading volume; $\alpha_o$, $\alpha_i$, $\phi_i$, $\gamma_o$, $\varphi_i$ are coefficients to be estimated; and $\varepsilon_t$ and $u_t$ are the error terms.

All estimations are implemented using the Stata software.

## 4. Results and Discussion

### 4.1. Descriptive Statistics

The descriptive statistics of the daily data are presented in Table 2 and Figure 1.

**Table 2.** Descriptive Statistics.

|  | Returns | Trading Volume | LN (Trading Volume) |
|---|---|---|---|
| Mean | −0.03566% | 661,681 | 12.35282 |
| Standard deviation | 0.73500% | 2,356,822 | 1.22595 |
| Minimum | −4.30915% | 12,166 | 9.40640 |
| Maximum | 3.54633% | 55,000,000 | 17.82284 |
| Skewness | 0.29756 | 12.63 | 0.75030 |
| Kurtosis | 6.346688 | 217.21 | 3.844766 |
| Chi$^2$ (LM test for the ARCH effect) | 76.398 *** | - | 52.588 *** |
| Nb. Observations | 1948 | 1949 | 1949 |

Note. *** indicates significance at the 1% level.

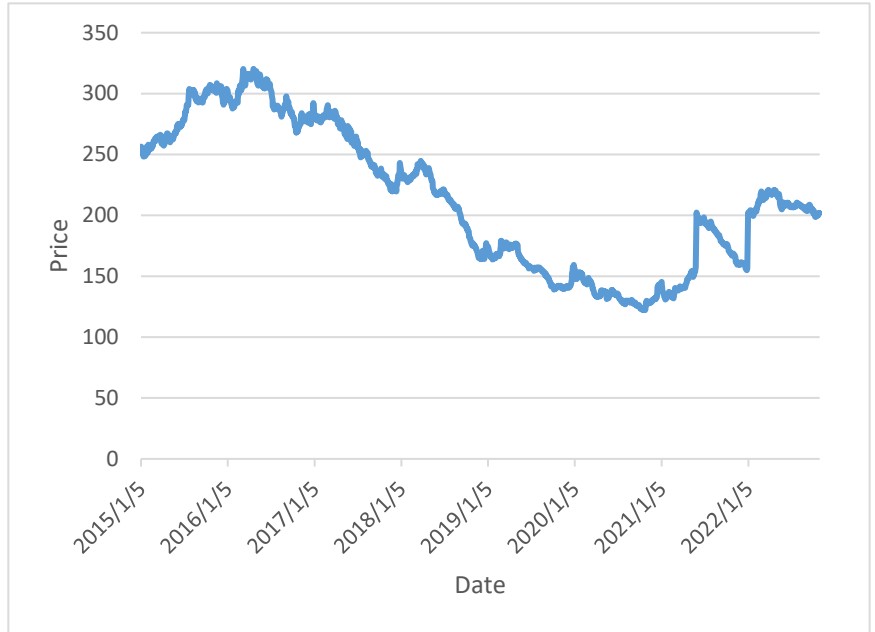

**Figure 1.** Stock index trend.

The average daily return is negative (−0.03566%) and highly volatile (0.7350% standard deviation). It ranges from −4.30915% to +3.54633%. The negative mean return is also corroborated by the decreasing stock price trend reported in Figure 1. The figure shows that in this market, the stock price has declined continuously from 5 January 2015 to 27 October 2020. It also shows that the BRVM market capitalization fell from 6276 billion XOF on 5 January 2015 to 3719 billion XOF on 27 October 2020 (a decrease of 40.74 %). The evolution of returns is reported in Figure A1 in the Appendix A.

It is clear at first glance that this trend does not offer investors attractive prospects for returns. The mean trading volume is 661,681 units and highly volatile (Figure 2) with a standard deviation of 2,356,822 units exchanged. The performance of the market during the years of analysis is bad. This is probably biasing the returns enormously.

The returns and the logarithm of the trading volume series have a positive skewness (0.29756 for returns and 0.75030 for volume) and excess kurtosis (6.346688 > 3 for returns and 3.844766 > 3 for volume). These values mean that the two series are not normally distributed. Furthermore, the autoregressive conditional heteroscedasticity (ARCH) tests indicate that the two series display some heteroscedasticity.

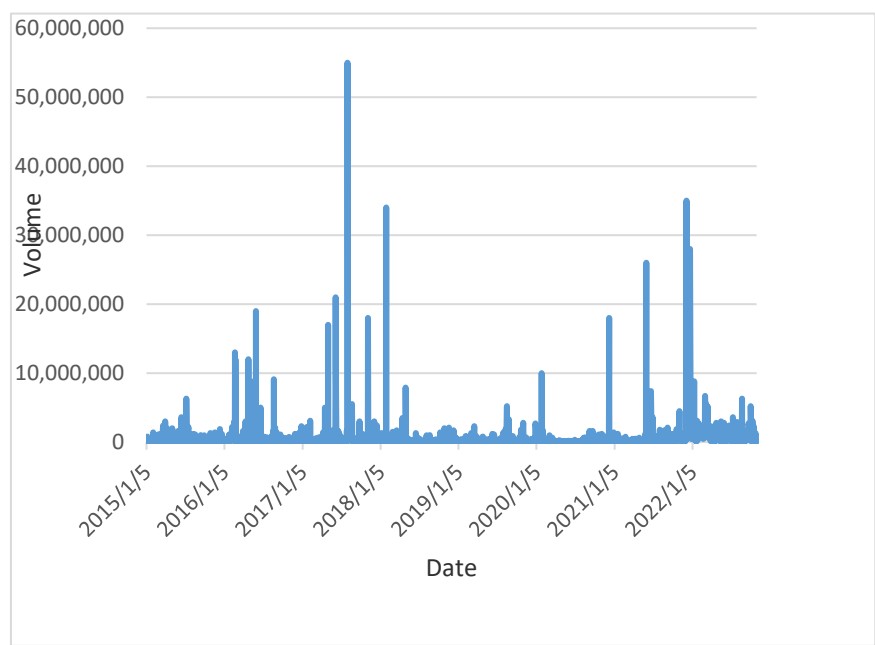

**Figure 2.** Evolution of the trading volume.

*4.2. Contemporaneous Results Using the GMM Specification*

Table 3 reports the results of the GMM estimation.

**Table 3.** GMM Estimation Results.

| $R_t = a_0 + a_1 V_t + a_2 R_{t-1} + a_3 V_{t-1} + \varepsilon_t$ <br> $V_t = b_0 + b_1 R_t + b_2 R_{t-1} + b_3 V_{t-1} + u_t$ | | |
|---|---|---|
| | **Returns** | **Volume** |
| Constant | −0.00179 <br> (−0.88) | 24.71416 <br> (0.63) |
| Return | - | 13,743.75 <br> (0.49) |
| Volume | 0.000073 <br> (0.49) | – |
| Return (−1) | −0.0307897 <br> (−0.95) | 423.1663 <br> (0.46) |
| Volume (−1) | 0.000043 <br> (0.33) | −0.59960 <br> (−0.24) |
| Number of observations | 1943 | 1943 |

Notes. The t of the student parameter is reported between parentheses. $R_t$ is the BRVM return at day t, and $V_t$ is the logarithm of the trading volume of day t.

The results in Table 3 show that returns and volume do not mutually influence one another. There is no contemporaneous impact of trading volume on returns. The coefficient of volume is positive (0.000073), but it is not significant. Regarding the impact of returns on trading volume, the coefficient is positive 13,743.75, but it is also insignificant. Thus, neither of the two variables influences the other.

*4.3. Contemporaneous Results Using the GARCH (1,1) Specifications*

Table 4 reports the results of the contemporaneous estimations using the GARCH (1,1) specification.

**Table 4.** Results for the GARCH (1,1) Specification.

| $R_{t=}\,a_0+a_1V_t+\varepsilon_t,\ \varepsilon_t\vert(\varepsilon_{t-1},\ \varepsilon_{t-2,\ldots\ldots})\to N(0,h_t)$ | | $V_{t=}\,b_0+b_1R_t+u_t,\ u_t\vert(u_{t-1},\ u_{t-2,\ldots\ldots})\to N(0,h_t)$ | |
|---|---|---|---|
| $h_{t=}\,\delta_0+\delta_1\varepsilon^2_{t-1}+\delta_2 h_{t-1}$ | | $h_{t=}\,\lambda_0+\lambda_1\varepsilon^2_{t-1}+\lambda_2 h_{t-1}$ | |
| | **Returns** | **Volume** | |
| Constant | −0.001744 (−1.12) | 12.27701 *** (41.82) | |
| Return | - | 1.457052 (0.42) | |
| Volume | 0.0002313 (0.80) | - | |
| Variance equation | | | |
| Constant | 0.00001 *** (8.08) | 0.110506 *** (2.98) | |
| ARCH | 0.155526 *** (8.87) | 0.068647 *** (4.53) | |
| GARCH | 0.673301 *** (20.61) | 0.856082 *** (23.51) | |

Notes. The t of the student parameter is reported between parentheses. *** indicates significance at the 1% level. $R_t$ is the BRVM return at day t. $V_t$ is the logarithm of the trading volume of day t.

The GARCH (1,1) results confirm the absence of any significant relationship between returns and trading volume. The coefficient of volume is positive (0.0002313), but it is not significant. For the impact of returns on trading volume, the coefficient is positive 1.457052, but it is also insignificant. Accordingly, neither of the two variables influences the other.

Thus, our contemporaneous results are robust to the estimation model used. Hypotheses 1a and 1b are then rejected. Neither of the two variables influences the other.

For both the return and volume equations, the ARCH and GARCH parameters are significant (at the 1% level for the returns, and at the 5% and 1% levels for the volume). This indicates that these series follow a heteroskedastic trend.

*4.4. Dynamic Results Using the Vector Auto Regression (VAR) Specification*

We started by determining the number of lags to include in the VAR estimation. Using different criteria, we set it at 5. The results are presented in Table 5.

**Table 5.** Optimal Number of Lags (in Days).

| | | **Criteria** | | | |
|---|---|---|---|---|---|
| | | **FPE** | **AIC** | **HQIC** | **SBIC** |
| | Lag | | | | |
| Returns | 0 | 0.000054 | −6.98447 | −6.98340 | **−6.98158 *** |
| | 1 | 0.000054 | −6.98433 | −6.98221 | −6.97857 |
| | 2 | 0.000054 | −6.98638 | −6.98320 | −6.97773 |
| | 3 | 0.000054 | −6.98583 | −6.98159 | −6.97430 |
| | 4 | 0.000054 | −6.98655 | −6.98125 | −6.97214 |
| | 5 | **0.000054 *** | **−6.99078 *** | **−6.98442 *** | −6.97349 |
| Volume | 0 | 1.50100 | 3.24401 | 3.24507 | 3.24688 |
| | 1 | 1.35873 | 3.14443 | 3.14654 | 3.15016 |
| | 2 | 1.30851 | 3.10677 | 3.10677 | 3.11537 |
| | 3 | 1.26888 | 3.07601 | 3.07601 | 3.08748 |
| | 4 | 1.24432 | 3.05646 | 3.05646 | 3.0708 |
| | 5 | **1.23725 *** | **3.05076 *** | **3.05709 *** | **3.06797 *** |

Notes. FPE = final prediction error; AIC = Akaike information criterion; HQIC = Hannan Quinn information criterion and SBIC = Schwartz information criterion. The bold and the * refer to the optimal number of lags determined by each criteria. All the criteria except for one (SBIC) suggest five lags for returns. All the criteria suggest five lags for volume.

Thus, the VAR model stands as follows:

$$R_t = \alpha_0 + \sum_{i=1}^{5} \alpha_i R_{t-i} + \sum_{i=1}^{5} \phi_i V_{t-i} + \varepsilon_t$$

$$V_t = \gamma_0 + \sum_{i=1}^{5} \gamma_i R_{t-i} + \sum_{i=1}^{5} \varphi_i V_{t-i} + u_t$$

Table 6 presents the results of the estimation.

**Table 6.** VAR Estimation Results.

|  | Returns | Volume |
| --- | --- | --- |
| Constant | −0.00218 (−0.84) | 4.620716 *** (11.86) |
| Return (−1) | −0.03354 (−1.48) | 6.97551 ** (2.04) |
| Return (−2) | 0.05271 ** (2.32) | 2.56761 (0.75) |
| Return (−3) | 0.01972 (0.87) | −0.18029 (−0.05) |
| Return (−4) | 0.04248 * (1.87) | 2.90875 (0.85) |
| Return (−5) | 0.07139 *** (3.14) | 7.88102 ** (2.30) |
| Volume (−1) | −0.000043 (−0.29) | 0.17873 *** (7.88) |
| Volume (−2) | −0.000051 (−0.33) | 0.11476 *** (5.03) |
| Volume (−3) | 0.000254 * (1.68) | 0.13056 *** (5.73) |
| Volume (−4) | 0.000017 (0.11) | 0.12286 *** (5.41) |
| Volume (−5) | −0.000024 (−0.16) | 0.07938 *** (3.53) |
| Number of observations | 1964 | 1964 |

Notes. The t of the student parameter is presented in parentheses. *, **, *** indicate significance at the 10%, 5% and 1% levels, respectively. $R_t$ is the BRVM return at day t. $V_t$ is the logarithm of the trading volume of day t.

Table 6 indicates that past returns (but not past volume) are significantly related to the current return series. Specifically, the second lag return is negatively and significantly (at the 5% level) related to the current return, the fourth lag is negatively and significantly (at the 10% level) related to the current return, and the fifth lag return is positively and significantly (at the 1% level) related to this current return. Conversely, except for a very marginal effect of the third lag (at the 10% level), none of the lagged volume is related to the current returns. Clearly, trading volume does not Granger-cause returns. Thus, Hypothesis 2b is rejected.

All the five past transaction volume are positively and significantly (at the 1% level) related to the current volume value. Conversely, only the first and fifth lags of returns are positively and significantly (at the 5% level) related to the current volume. The Granger causality test shows that stock returns Granger-cause volume. Therefore, in the BRVM stock market, there is a unidirectional causality running from stock returns to trading volume. We cannot reject Hypothesis 2a.

Table 7 presents the Granger causality Wald test results.

**Table 7.** Results of the Granger Causality Wald Tests.

| Excluded | Chi$^2$ | DF | Prob > Chi$^2$ |
|---|---|---|---|
| Panel A: Dependent variable: Stock returns | | | |
| Logarithm trading volume | 3.0485 | 5 | 0.693 |
| All | 3.0485 | 5 | 0.693 |
| Panel B: Dependent variable: Logarithm trading volume | | | |
| Stock returns | 11.144 | 5 | 0.049 |
| All | 11.144 | 5 | 0.049 |

From Table 7, Panel A, we observe that the null hypothesis that the logarithm of the trading volume does not Granger-cause stock returns should not be rejected at the regular 5% level, as the associated *p*-value is as high as 0.693 (above 0.05). Conversely, from Panel B, we see that the null hypothesis that stock returns do not Granger-cause the logarithm of the trading volume should be rejected at the regular 5% level, as the associated p-value is as low as 0.049 (below 0.05).

*4.5. Results Discussion*

The results presented above show that contemporaneous stock returns and trading volume do not mutually influence one another. Thus, there is no significant mutual contemporaneous relationship between the two series in the BRVM. This result is contrary to the result of Mpofu (2012), who reports a significant relationship between the absolute value of price change and trading volume in the South African stock exchange. Similarly, it is contrary to the findings of Lee and Rui (2002), who report a positive contemporaneous relationship between returns and trading volume in the USA, Japan and the UK. This result shows that the MDH does not apply to the BRVM since the expected positive correlation is not present.

The dynamic specification is estimated through the vector autoregressive model. It shows that past stock returns are positively related to the current of trading volume. This establishes a causality running from stock returns to trading volume. Ngene and Mungai (2022) have reported a similar result for six African stock exchanges: Egypt, Kenya, Morocco, Nigeria, South Africa and Tunisia (but not for Ghana and Mauritius). For the second part of the dynamic specification, trading volume does not Granger-cause stock returns. This result implies that the information contained in the stock trading volume cannot significantly improve the ability to predict the stock price. The same conclusion is reached by Habib (2011) for the Egyptian stock exchange over the period 1998–2005; by Akpansung and Gidigbi (2015) for the Nigerian stock exchange over the period 1981-2012; and by Lee and Rui (2002) for stock markets in the US, Japan and the UK.

The absence of causality from trading volume to stock returns invalidates the SIAH for the BRVM. Once MDH and SIAH are invalidated, the dynamic behind the data remains to be determined. Based on past returns (the second and the fifth lags), investors increase their transaction volume. What is the logic behind this? Do investors trade based on behavioral attitude such as overconfidence? Some theories argue that high returns make investors overconfident, and consequently, they trade more (Glaser and Weber 2009). Further deep analysis is necessary to determine if this is the case here.

Our results nuance the observations of Essingone and Diallo (2022) that "The BRVM [ . . . ] market is essentially composed of 'fundamentalist' investors who buy securities and hold them in order to receive dividends. This investors' profile contrasts with that commonly observed in developed stock markets, which are mainly interested in returns and potential capital gains. Thus, investors in BRVM seem to take little or no risk and are only interested in stocks that have potential and/or 'visibility' with a high current yield.".

## 5. Conclusions

The objective of this paper was to study the contemporaneous and dynamic relationship between the return on securities and the trading volume in a developing market: the BRVM. Using daily data from the BRVM stock index over the period 5 January 2015 to 31 October 2022 (a total of seven years, ten months or 1949 observations), returns were regressed on the logarithm of volume, and vice-versa. The statistical evidence indicates that there is no contemporaneous significant relationship between stock returns and trading volume. This result shows that the MDH does not apply to the BRVM since the expected positive correlation is not present. Furthermore, the dynamic (the VAR model) was estimated for testing the casual relationship between stock returns and trading volume. The results indicate a unidirectional causality running from stock returns to trading volume.

The implication of the absence of a volume to price changes (returns) causality is that, at the BRVM, changes is prices are not driven by large unexpected transaction volume, that exceed the market normal size. This absence of causality may reflect usual low trading volume. Large trading volume usually occur only after past positive returns (thus the causality from returns to volume).

Following these results, the recommendation (policy implication) is that at the BRVM, actions and policies undertaken to revitalize the market must first target returns, before volumes.

While the results found in this paper are consistent with the findings of earlier studies, the paper has some limitations. First, the period covered runs from 2015 to 2022. It could be extended to include periods such as the 2008 financial crisis and gather the relationship during crisis periods. Second, only linear methods have been used here. Additional methods such as non-parametric causality models could be considered. Third, the analyses here are limited to the BRVM index. Will individual stocks, or analyses by industry provide more insights?

Thus, it is recommended that further studies be conducted at the firm (i.e., on individual stocks) as well as at industry levels. It is also recommended that the analysis be conducted by type of investors (e.g., informed versus uninformed) to enhance a better understanding of the BRVM securities exchange. For instance, the use of heterogeneous-agent trading models can determine how trading volume reflects the quality of traders' information signals and help to disentangle whether returns are associated with portfolio-rebalancing trades or information motivated trades (Gagnon and Karolyi 2009). Non-parametric causality models should be considered for a robustness check. Finally, deep analysis is needed for a better understanding of investors' dynamics at the BRVM.

Addressing the issue of trading volume at the BRVM stock exchange necessarily brings to mind other angles of analysis that have not been covered in this article but which deserve future attention. This is the case for herding behavior, liquidity synchronization and the transmission mechanisms of foreign direct investments (FDI) to economic sectors. Ukpong et al. (2021) have examined the herding in the US market using both market and industry level data over the period 1990–2020. They found that herding does not exist at the market level, but becomes visible at the industry level. Zaidi and Rupeika-Apoga (2021) have addressed the issue of liquidity synchronization (i.e., the impact of market-wide liquidity changes on individual stock liquidity) and its determinants on seven emerging Asian markets (Bangladesh, China, India, Indonesia, Malaysia, Pakistan and the Philippines) over the period 2010–2019. They found a strong evidence of liquidity synchronization in these countries. Pečarić et al. (2021) have analysed the transmission mechanisms of the sectoral FDI inflows on a sample of ten Central and East European countries. This analysis helps identify how FDI inflows (which are part of trading volume) are channelled across different sectors within an economy, depending on the levels of keys economic indicators. Assuming that data are easily accessible, such analyses could also be undertaken for the BRVM.[6]

**Author Contributions:** Conceptualization, J.-P.G. and M.S.D.; methodology, J.-P.G. and M.S.D.; software, J.-P.G. and M.F.D.; validation, J.-P.G., M.S.D. and M.F.D.; formal analysis, M.S.D. and

M.F.D.; data curation, M.S.D. and M.F.D.; writing—original draft preparation, J.-P.G. and M.S.D.; writing—review and editing, J.-P.G., M.S.D. and M.F.D.; supervision, J.-P.G.; project administration, M.S.D. All authors have read and agreed to the published version of the manuscript.

**Funding:** This research received no external funding.

**Informed Consent Statement:** Not applicable.

**Data Availability Statement:** The datasets used and/or analyzed during the present study are available from the corresponding author upon reasonable request.

**Conflicts of Interest:** The authors declare no conflict of interest.

## Appendix A

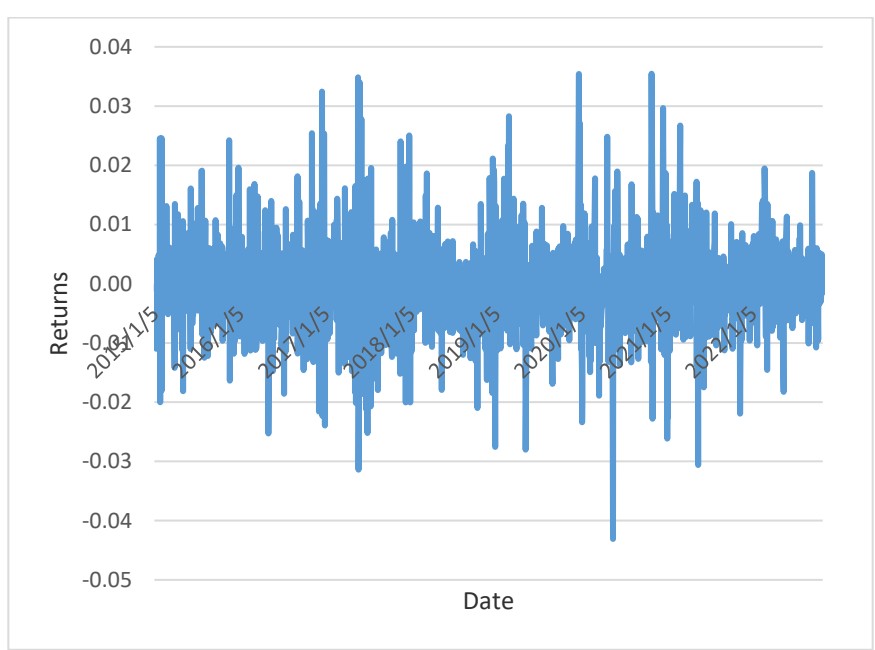

**Figure A1.** Evolution of daily index returns.

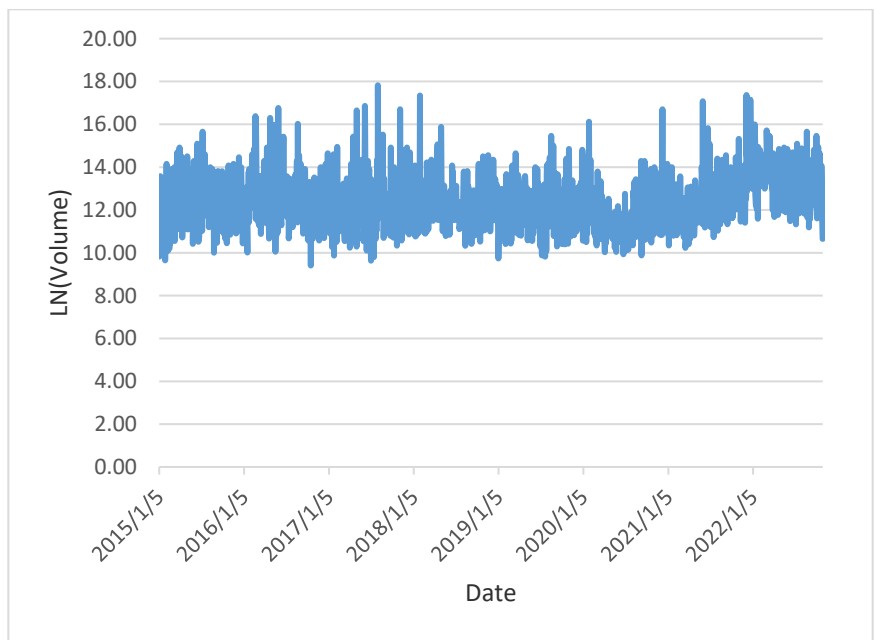

**Figure A2.** Evolution of the logarithm of the trading volume.

## Notes

[1]   BRVM is a regional stock exchange serving the following West African Economic and Monetary Unit (WAEMU) countries: Benin, Bissau Guinea, Burkina Faso, Ivory Coast, Mali, Niger, Senegal and Togo. It was created on 18 December 1996 in accordance with a decision of the WAEMU Council of Ministers taken in December 1993. It is based in Abidjan (Ivory Coast) and has national stock exchange branches in each of the member countries. The effective start of its activities took place in September 1998. BRVM was created as the result of a long process of institutional, technical, political and economic integration of WAEMU countries. It was set up in several stages (https://www.brvm.org/fr/historique): accessed on 7 November 2022.

-   14 November 1973: signing of the treaty establishing the West African Monetary Union (WAMU) comprising Benin, Burkina, Côte d'Ivoire, Mali, Niger, Senegal and Togo. Guinea-Bissau was added to the treaty in 1997;
-   17 December 1993: decision by the WAMU Council of Ministers to create a regional financial market and to mandate the Central Bank of West African States (BCEAO) to conduct the project;
-   18 December 1996: creation of BRVM and DC/BR as private limited companies in Cotonou, Benin;
-   20 November 1997: installation by the WAMU Council of Ministers of the Regional Council for Public Savings and Financial Markets (CREPMF), a market regulation body;
-   16 September 1998: start of BRVM and DC/BR activities;
-   24 March 1999: start of Decentralised Electronic Listing;
-   12 November 2001: switch to daily listing;
-   2 July 2007: changeover from T + 5 to T + 3 (DC/BR) settlement/delivery time;
-   16 September 2013: switch to continuous listing (BRVM);
-   19 March 2018: opening of the 3rd share compartment of the BRVM.

   BRVM ranks sixth among African stock exchanges behind South Africa, Nigeria, Morocco, Egypt and Kenya. It represents 10% of the gross domestic product of the WAEMU. The market capitalisation of the equity market has risen from XOF 1108 billion in 1998 to XOF 4740.6 billion on 31 December 2019, an increase of 327% since the start of the Exchange's activities. In twenty three (23) years of operation, the BRVM has traded 889 million securities with a total value of more than XOF 2285 billion.

[2]   As noted by Ngene and Mungai (2022), "The continuous and time-honored rationale for the empirical research on the stock return-volume and volume-volatility causal dynamics has been driven by the yearning to test the validity of two Wall Street adages: (i) It takes volume to make prices move. This adage implies that volume causes return and that there is a positive relationship between volume and absolute (magnitude) measure of returns or volatility. (ii) Volume is relatively in bear markets. This adage implies that returns cause volume and that there is a positive co-movement between volume and return.". Furthermore, "The rate of profit is one of the most important indicators for the stakeholders and shareholders of the companies in the modern economy" (Dospinescu and Dospinescu 2019). Additionally, understanding the dynamics behind stock returns and trading volume is also very important for them.

[3]   The correlation between lagged stock returns and current trading volume can be derived from behavioral finance models. Investors' transactions on financial markets (and consequently trading volume) are not always driven by rational expectations. Behavioural attitude such as overconfidence may intervene (Odean 1998, 1999; Gervais and Odean 2001; Hirshleifer and Luo 2001; Baker and Nofsinger 2002; Statman et al. 2006; Glaser and Weber 2009). The work of Kahneman and Tversky (1979) considers that the psychology of the investor is one of the factors to be taken into consideration in order to understand how investment decisions are influenced in a risky environment such as the stock market.

[4]   At the BRVM, two main indices represent the activity of the equity securities market: the BRVM Composite and the BRVM 10. The BRVM Composite is a market capitalization-weighted index that tracks the daily total return performance of all companies listed on the BRVM. It is adjusted each time a new company is listed (but also in the event of a capital increase or reduction), so as to be adapted to changes in the Regional Financial Market. It is therefore very sensitive to the capitalization of large stocks. The BRVM 10 is made up of the ten most active companies (or more simply, the ten most liquid companies) on the market. These are listed companies whose shares have been bought or sold the most. The list is updated four times per year. BRVM indices quotes are automatically generated by the BRVM trading system and are disseminated after each trading session.

[5]   For the contemporaneous regression estimation, GMM and GARCH models are preferred over the simple OLS model, because they are more suitable to deal with heteroscedasticity, endogeneity and simultaneity biases. They produce heteroscedasticity-consistent estimates (Lee and Rui 2002).

[6]   We thank an anonymous referee for suggesting this addition.

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
