# Peer review of "Relationship between Stock Returns and Trading Volume at the Bourse Régionale des Valeurs Mobilières, West Africa"

_ijfs, doi:10.3390/ijfs10040113_

Round 1

Reviewer 1 Report

I read the paper and have some comments for improvements of the paper as follows:

1. The motivation of the paper is very weak. The topic of the paper has been widely documented in the literature. I find nothing new in the manuscript.

2. I find some papers that are related to the topic of the paper in Africa, but they have not been included in the literature review of the paper.

3.  There are some concerns in the research methodology of the study as follows:

-       Why are the GMM and GARCH(1,1) methods employed in the study?

-       What are differences between GMM, GARCH(1,1) and VAR models that are used in the study?

-       The authors should explain how they selected the instrument variable for the GMM model.

4. Results of Granger test should be added in the manuscript.

The authors should pay more attention to presenting their manuscript. There are some mistakes in the paper.

-       Some sections are incorrectly numbered.

-       The conclusion is duplicated.

5. Some recommendations/implications for stakeholders should be included in the conclusion section.

Reviewer 2 Report

Please see the attached file for a detailed comments. 

Reviewer 3 Report

Dear Author(s),

Please find below my concerns and recommendations regarding your manuscript proposal entitled "Relationship between Stock Returns and Trading Volume at the Bourse Régionale des Valeurs Mobilières, West Africa" sent to International Journal of Financial Studies.

I have read the Introduction section and I found the research gap and the research goal, but I recommend you to also clearly define the research question, so that the readers what your article wants to answer to.

The Literature Review section must be improved from many points of view.

First of all, at the end of this section you should define and describe the research hypotheses. Every modern scientific article must define at least one research hypothesis that should be tested within the article.

Secondly, the resources used in Lit. Review are pretty old. I recommend you to enhance the general context of your research proposal and cite here the following valuable resources: https://doi.org/10.1016/j.frl.2021.101953, http://www.ecoforumjournal.ro/index.php/eco/article/view/884, https://doi.org/10.3390/risks9020043, https://doi.org/10.3390/economies9020066.

At rows 117-119 you say: "The data used in this paper consist in daily market price index and daily trading volume series for the Bourse Régionale des Valeurs Mobilières (BRVM). They are extracted from the BRVM financial database. They cover the period of 05 January 2015 to 31 May 2021, a total of six years or 1573 observations."

Please argue why you chosed this period and why it is relevant for your research proposal.

Can be extrapolated the results from that period?

In table 6 (Table 6. VAR estimation results), between rows 210-211, you have the values for Returns and Values. Some values are presented in bold style. Why? Should you present additional information about that values? Or is only an editing issue?

Please revise this aspect.

At rows 241-242 you say: "The BRVM […] market is essentially composed of ‘fundamentalist’ investors who but securities and hold them in order to receive dividends."

There is a mistake in the text: instead of "...who but securities..." it should be "...who BUY securities...".

Please revise and correct this issue.

In the current text, you have two "Conclusion" chapters.

The first one is at row 246, and the second at row 263.

Please revise and correct it. 

Dear Author(s),

Please consider all the above recommendations as being constructive remarks in order to improve the general quality of your manuscript proposal.

Kind Regards!

Round 2

Reviewer 1 Report

All my comments have been considered for revising.  The efforts of the authors are highly appreciated.

Author Response

Thank you for your kind suggestion.

Reviewer 3 Report

Dear Author(s),

I have read your answers to the recommendations from the previous round of review and I also analyzed the new version of the manuscript proposal.

Please find below my concerns about your document:

1. During my documentation for this review, I found that in the new version of the article, many sequences of texts are similar to the others already published in some articles. For example, the rows 54-67 are similar to the source available at https://doi.org/10.1016/j.irfa.2022.102176.  The same situation for the paragraph from page 19, rows 4-9.

The rows 147-154, 165-176 are vey similar to https://doi.org/10.1080/17520843.2021.1953865. 

The rows 107-116 are very similar to https://ueaeprints.uea.ac.uk/id/eprint/61510/1/Accepted_manuscript.pdf.

And so on...

Please revise the whole document from this point of view and revise the texts.

Regarding the recommended references from the previous round of review, their role is to widen the general context of your research proposal and I still recommend you to include them in your manuscript.

At rows 400-402 you say: "Trading activity and market capitalisation are heavily concentrated with SONATEL, the Senegalese telecommunications company, which accounts for 53.95% of the market value and 46.51% of the capitalisation (Hearn and Piesse 2010)."

My question is if a reference from 2010 (about 12 years ago) is still valid today, when you are conducting this research. Please check the liquidity of the analyzed stock exchange.

Kind Regards!

Round 3

Reviewer 3 Report

Dear Author(s),

After reading the revised version of the manuscript,  consider that you successfully addressed all my constructive recommendations from the previous round of review.

Kind Regards!

Author Response

Thank you for your kind suggestion